# An Integrated Approach for Formation Micro-Image Rock Typing Based on Petrography Data: A Case Study in Shallow Marine Carbonates

**Polina Kharitontseva** [1,*] **, Andy Gardiner** [1,2] **, Marina Tugarova** [3] **, Dmitrii Chernov** [1,4] **, Elizaveta Maksimova** [3] **, Ilia Churochkin** [1] **and Valeriy Rukavishnikov** [1]

1   School of Earth Sciences & Engineering, Tomsk Polytechnic University, 634050 Tomsk, Russia; Andy.Gardiner@hw.ac.uk (A.G.); chernovda95@mail.ru (D.C.); ChurochkinII@hw.tpu.ru (I.C.); RukavishnikovVS@hw.tpu.ru (V.R.)

2   Institute of Geoenergy Engineering, Heriot-Watt University, Edinburgh EH14 4AS, UK

3   Gazpromneft Science & Technology Center, 190000 St. Petersburg, Russia; Tugarova.MA@gazpromneft-ntc.ru (M.T.); Maksimova.EN@gazpromneft-ntc.ru (E.M.)

4   SAMARANIPINEFT, 443010 Samara, Russia

*   Correspondence: kharitontsevapa@hw.tpu.ru

**Abstract:** Core rock-typing (RT) is commonly used for creating geologically reliable models of porous media in carbonate reservoirs. This approach is more advanced than the traditional porosity–permeability relationship and is based on the division of carbonate rocks into groups, using common classifications (lithofacies, FZI, Winland–Pittman, etc.). These clustering methods can provide either geological or petrophysical descriptions of the identified rock types. Besides, the connection of identified core rock types with standard logs could be challenging due to the different scales of measurement. This paper considers the creation of a new approach, named "integrated rock-typing," which connects geologically and petrophysically driven rock types using borehole image logs. The methodology was applied to an Upper Devonian–Lower Carboniferous carbonate field. The workflow comprises borehole image structural/textural analysis with vug fraction identification, quantitative geological descriptions from thin sections, and petrophysical measurements. The geological section is divided into six rock types, which were controlled by sedimentary and diagenetic processes. The created Rock Type Catalogue provides clear links between rock types and log data, including wells with standard suites of logs. The results will be useful for geological modelling and validation of the future drilling strategy for the studied field.

**Keywords:** rock typing; carbonate reservoirs; petrography; FMI log; porosity spectrum analysis; data matching

## 1. Introduction

Carbonate reservoir rocks are primarily represented by limestones and dolomites and characterized by complex pore networks, which can be created by a combination of different grain types, grain textures, mineral compositions, and diagenetic processes. A sedimentary environment determines the initial porosity of carbonate rocks. Later diagenetic processes and tectonic movements may have a significant influence on porosity and permeability by the formation of secondary porosity, including fractures, molds, and vugs. The dissolution of calcite can significantly increase the porosity in limestones and open natural fractures can create a connected system in low-porosity carbonates, allowing hydrocarbons to flow [1]. Since a significant proportion of pores in carbonates, particularly secondary pores, may be isolated from each other, the permeability of carbonate layers can vary by several orders of magnitude for the same porosity. Poor correlation between reservoir porosity and permeability caused by its strong vertical and lateral heterogeneity, characterizing the carbonate reservoir and building the permeability model,

is challenged [2]. Consequently, standard permeability models based on $k$–$\varphi$ (permeability–porosity) relationships often do not support the creation of reliable geological models to be used for reservoir property predictions.

Rock types could be subdivided into clusters by using various automation algorithms, such as k-means cluster analysis, model-based, and hierarchical clustering methods. The main idea is using the continuous data, such as well log data (gamma ray, neutron, acoustic, density, induction, and resistivity logs) from wells to identify similar features that could be assigned as rock type in the stratigraphic column. The k-means algorithm partitions the input data set into k partitions (clusters) by using k observations to serve as centers for the k-clusters. Then, the distance from each of the other observations is calculated for each of the k-clusters, and observations are put in the closest clusters. That process is repeated until no observations switch clusters [3,4]. Model-based clustering uses the expectation-maximization (EM) algorithm to fit a mixture of multivariate normal distributions to a data set by maximum likelihood estimation [5]. Hierarchical cluster analysis is an algorithm that groups similar objects into groups called clusters. The endpoint is a set of clusters where each cluster is distinct from all the other clusters and the objects within each cluster are broadly similar to each other. Hierarchical clustering starts by treating each observation as a separate cluster. Then, it repeatedly executes the following two steps: (1) Identify the two clusters that are closest together, and (2) merge the two most similar clusters. This iterative process continues until all the clusters are merged together [6].

The latest research shows an increasing interest in these rock-typing approaches [7–10]; however, in terms of carbonates, questions still arise, as the complex pore space of the carbonate reservoir is often associated with secondary porosity, the appearance of which is facilitated by processes such as leaching, dolomitization, recrystallization, and fracturing. Due to these processes, the primary matrix is transformed, and the reservoir properties are no longer controlled by the lithofacies [11].

Therefore, it is extremely important to perform the rock-typing step before any static geomodelling of carbonate reservoirs. The typing of reservoir rocks (rock-typing, RT) is a very time-consuming process that requires a high level of professional expertise to integrate a large amount of data with different scales. The dataset generally includes the results of core and thin section description, routine core analysis (RCAL), special core analysis (SCAL), and standard and advanced logs, for example, electric borehole images.

The rock-typing process involves the division of rocks into groups or clusters. Usually, each rock type should have specific geological and petrophysical characteristics for it to be propagated in 3D geological and simulation models using clear geostatistical behavior [3,12]. However, the complex pore structure of carbonate reservoir rocks significantly affects the clear allocation of flow units due to several factors, including core-to-log upscaling and the availability of core and log data for all wells. Rebelle et al. [13] were able to identify the following problems regarding the rock-typing approach:

- Geologically driven rock types: The depositional facies, more or less impacted by diagenesis, are the main parameter used for discriminating the rock types. In this case, the texture (e.g., Dunham classification) and type of grains of the depositional facies can be used as parameters for classification [14–17]. The problems involved in geologically driven rock types are poor petrophysical discrimination and an uncertain link to the simulation model because core reservoir properties may be similar for different facies.
- Petrophysically driven rock types: Clustering is conducted based only on the petrophysical properties of the rocks. These properties are derived through conventional measurements (e.g., porosity, permeability, and grain density) or capillary pressure, and pore throat radius [18], the flow-zone indicator (FZI), the reservoir quality index (RQI), the Winland–Pittman R35 method, the Thomeer function, or rock clustering using machine learning [19–24]. However, petrophysically driven rock types can be allocated with too many lithotypes because of insufficient core sampling or a few advanced logs in exploration or production wells. Therefore, petrophysical rock

types could not be clearly distributed in a 3D geological model due to the absence of geostatistical trends.

- Production-driven rock types: Large-scale dynamic data (e.g., production logging tool (PLT) and dynamic rock types (DRT)) are the main parameters considered for this approach. Production-driven rock types have poor links with geological and petrophysical data because of the different scale of measurements. For example, core porosity and permeability are obtained from 30 mm core plugs, log vertical resolution is about 0.1–1 m, and production logging and well tests are used to analyze larger intervals, 10 m and higher [13,25,26].

To create reliable geological models for carbonates, the above-mentioned rock types should be integrated. This step is becoming more important for fields with a few wells in which core and borehole images are available. In this case, rock types can be clearly identified with core and image logs in key exploration wells. The main challenge will be the implementation of defined rock types in production wells with a limited number of logs.

The integrated rock-typing approach proposed in this article can address the above-mentioned challenges by likening geologically driven rock types, defined with a thin section description and petrography analysis, with petrophysically driven rock types, defined with borehole image analysis and well-log interpretation. The major advantage of the proposed integrated rock-typing approach is its applicability to wells without core. An integrated rock type combines knowledge of geology and petrophysics, and finds the relationship between identified rock types with conventional logging by estimated limits of each rock type in GR, RHOD, DT, LLD, and Neutron. The proposed approach differs from other conventional approaches by creating a catalogue of rock types that describes each rock type by means of FMI images (Static and Dynamic), core photos (DL, UV), thin section photos, poro-perm relationships, vug fraction, typical values of conventional well logs, and distinctive features. This data (part of data) could automatically be compiled from new wells and the established rock type could be found for each interval through the use of phyton scripts.

## 2. Materials and Methods

The present study considers the integration of geological and petrophysical data to improve reservoir characterization for a real carbonate field with limited availability of core and logs. The methodology includes quantitative geological descriptions obtained from the analysis of thin sections, as well as petrophysical measurements from log data (i.e., borehole image logs, standard logs).

The proposed approach requires the performance of two steps: (1) the study of data and determination of geological and petrophysical features of the reservoir based on core and image logs, and (2) integrated rock-typing—finding the link between core, borehole images, and standard logs. Figure 1 shows the correspondent workflow.

The main idea of the workflow is based on the use of formation microimagers (FMI) as a link between geology and petrophysics. On the one hand, 5 mm vertical resolution provides a direct comparison of core and FMI, which helps to identify structural and textural geological features of the reservoir. On the other hand, electrical borehole images are based on certain physical measurements of rock properties, e.g., electrical conductivity. These physical properties can be compared with other log data, especially for describing intervals that were not cored [27].

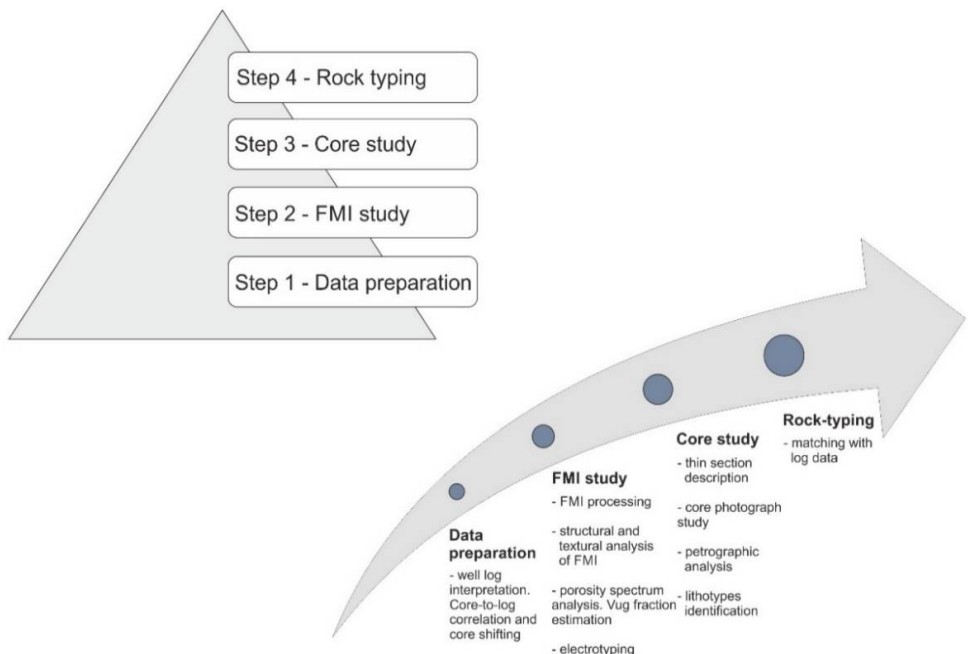

**Figure 1.** Overview of the rock-typing process. The pyramid shape indicates the amount of time spent on each step of the workflow. The first two steps are the most time-consuming, due to the great number of routine procedures.

Before starting the workflow, a set of pre-processing procedures should be carried out:

1. Core description, well-log interpretation, and core-to-log correlation;
2. Wellbore image processing or FMI processing;
3. Structural and textural analyses of the FMI;
4. FMI electrotype identification;
5. Quantitative analysis of vug fraction (PoroSpect method);
6. Thin section description (qualitative parameters);
7. Study of core photos;
8. Petrography analysis (quantitative parameters).

Among these procedures, the most time-consuming is the FMI processing and interpretation [28]. This includes quality control of the inclinometer, speed correction, the creation of a unified data array, and the normalization of electrode responses. The structural and textural analysis of the borehole images includes the identification of the reservoir boundaries, fractures, and stylolites.

The main disadvantage of the approach is the mandatory presence of a borehole image, at least one in the field or study area. If the conditions do not allow the borehole microimager to be recorded, or its interpretation is unsatisfactory, the approach is not applicable.

The main advantage of the approach is the possibility of using it in new wells, without core: rock types distinguished by geology (using the description of core and thin sections), and rock types distinguished by borehole imagers—these imagers are tied to a conventional set of logs through electrical conductivity. If necessary, convolution could be performed (two similar geology and petrophysics rock-types with different borehole image patterns could be combined into one), markers (characteristic values, peaks) on a standard logging complex could be found, and then supervised training could be conducted on newly drilled wells.

## 3. Case Study

### 3.1. Available Data and Geological Background

The N field is located in the Orenburg region (Figure 2) of Russia at a depth of 2475 m. The N field consists of three main carbonate reservoirs, C1t, D3zv, and D3fm, of Upper Devonian–Lower Carboniferous strata. The D3zv interval of interest was subdivided into layers Zl1, Zl2, and Df1.The research was based on six wells including structural maps, seismic sections, petrographic data, standard well logs, RCAL, and SCAL for all wells, and FMI only for four wells. To test the Python code, the data was divided into 2 sets (4 training wells and 2 test wells).

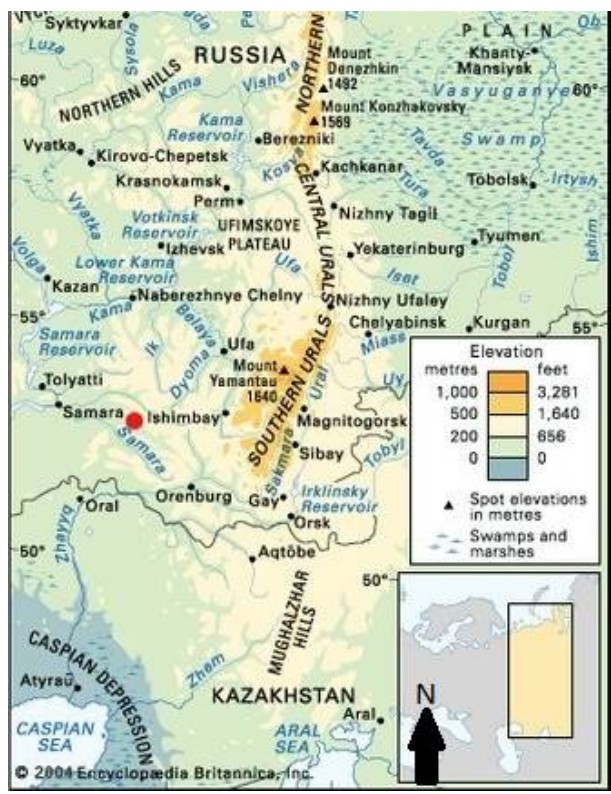

**Figure 2.** Schematic map showing the location of the area studied (red point—schematic field location), after Yastrebov, V. and Poulsen, T. M., Encyclopedia Britannica, Inc. (London, UK), [29].

The average thickness varied from 140 to 150 m. The permeability of the field varied from 0.1 to 10 mD and porosity varied from 4 to 8% [30]. The reservoir consists of light grey, bioclastic-detrital, and algal porous, layered limestones formed in a shallow marine environment, mostly in the bioherm shelf zone and shelf zone [31] (see Figure 3). The deposits are characterized by frequent vertical and lateral alternations of lithotypes and petrophysical properties. The reservoir primarily contains oil; however, in its predevelopment stage, it was believed to have included also an oil–water contact.

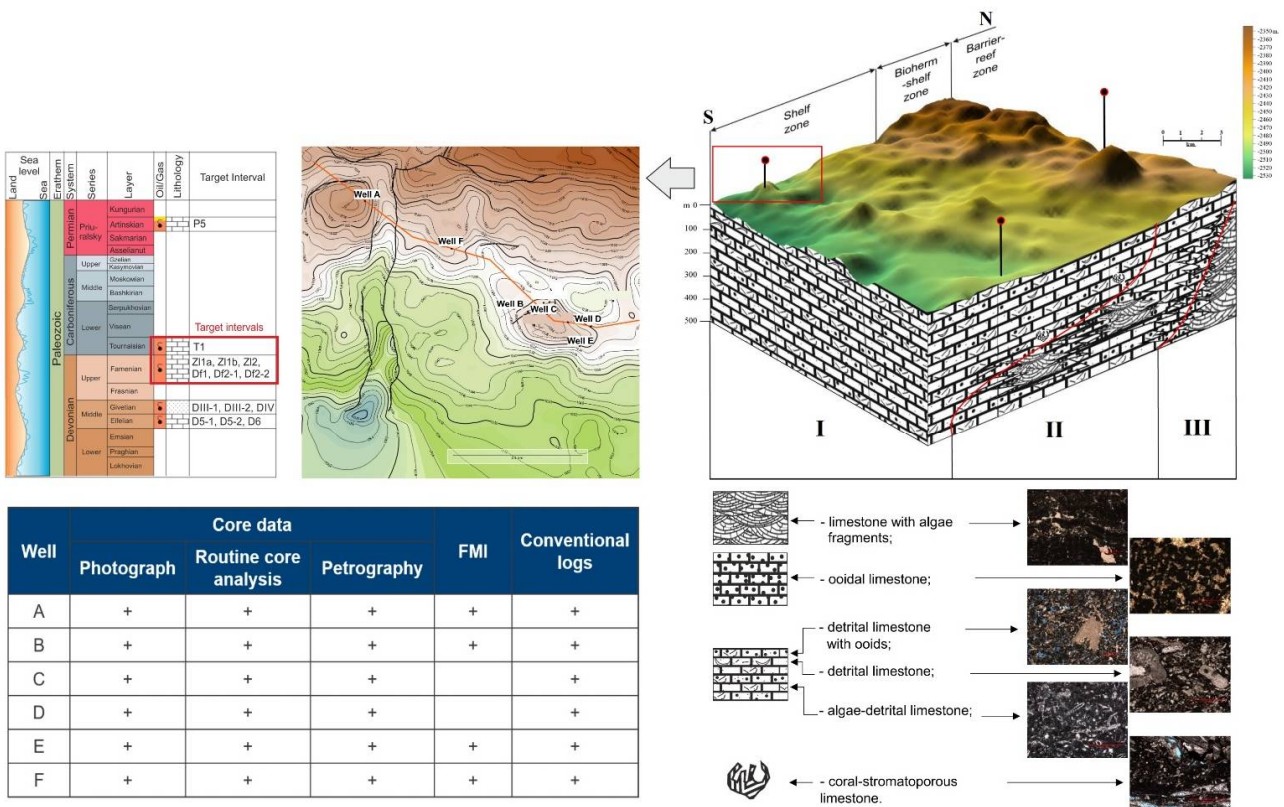

*Rostovtseva and Chekan et al., 2020*

**Figure 3.** Geological background of the studied field. The conceptual model was previously created based on the limited number of drilled exploration wells. Wells with FMI data were not used in the conceptual model. During the late Devonian time, three facial zones were formed: I—shelf zone, II—bioherm zone, and III—barrier reef zone. It happened when the late Famennian barrier reef stopped migrating.

Previous studies of the N field pointed to sedimentary features of the rocks having the main influence on the petrophysical properties. Diagenetic processes such as recrystallization, dissolution, and tectonic microfractures emphasize the facies heterogeneity, which adds complexity to log interpretation [30]. Consequently, the final porosity and permeability had non-uniform distribution trends and low values. However, such onshore reservoirs can still be economically viable for Russian oil companies. Under such conditions, a good reservoir model is even more important than usual.

### 3.2. Data Analysis

The workflow began with the pre-processing step: core description, well-log interpretation, core-to-log correlation, and core shifting. Log templates were prepared to visualize the study section. They included petrophysical data (including porosity and permeability curves and ultraviolet core photos), which were matched to the oil-saturated layers (see Figure 4).

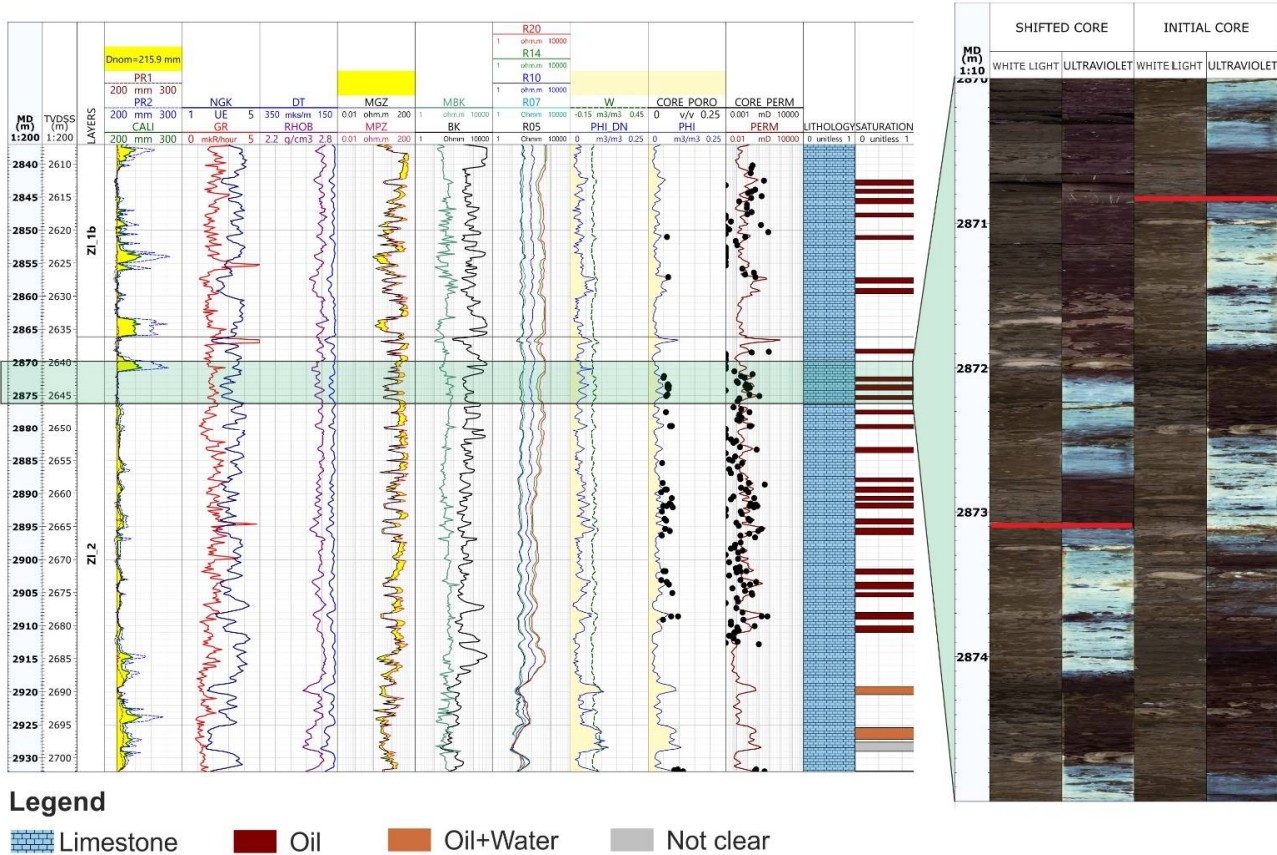

**Figure 4.** Well-logging interpretation and core shifting. Core was shifted using the comparison of core and log gamma ray measurements. This method provides an accurate estimation of shifting intervals. Continuous porosity and permeability curves are obtained from previous log interpretation. (CALI, PR1, PR2—calipers; NGK—neutron gamma log, MBK, MGZ, MPZ—microlaterologs, BK—laterolog; R05-R20—high-frequency electrical logging).

The next step included FMI data analysis. Micro-resistivity imaging (FMI), known as a wireline logging method, is based on the dipmeter principle and produces a high-resolution resistivity image of the borehole wall. The FMI consists of four or eight articulated pads containing two sets of electrodes in each pad. Each electrode emits an electric current to the wall of the well and each receiver button records a signal, from which the resistivity can be calculated. The buttons are slightly offset horizontally on the pads, such that each button records a high-resolution resistivity log from a narrow strip in the interior of the borehole. These multiple logs can then be combined to build up an image of the electrical resistivity/conductivity of the borehole wall. The FMI high vertical resolution (~5 mm) allows for the recognition of rock textural and structural features along the borehole wall, which can be visible on core or core photographs but usually cannot be recognized on any conventional wireline logs.

The FMI processing workflow combined several processing algorithms into a common sequence, which included the following steps:

1. Quality control of the inclinometer;
2. Correction for variable speed;
3. Creation of a unified data array;
4. Normalization of the electrode responses.

FMI processing is one of the main parts involved in an FMI study and allows the data to be presented in different ways (Figure 5).

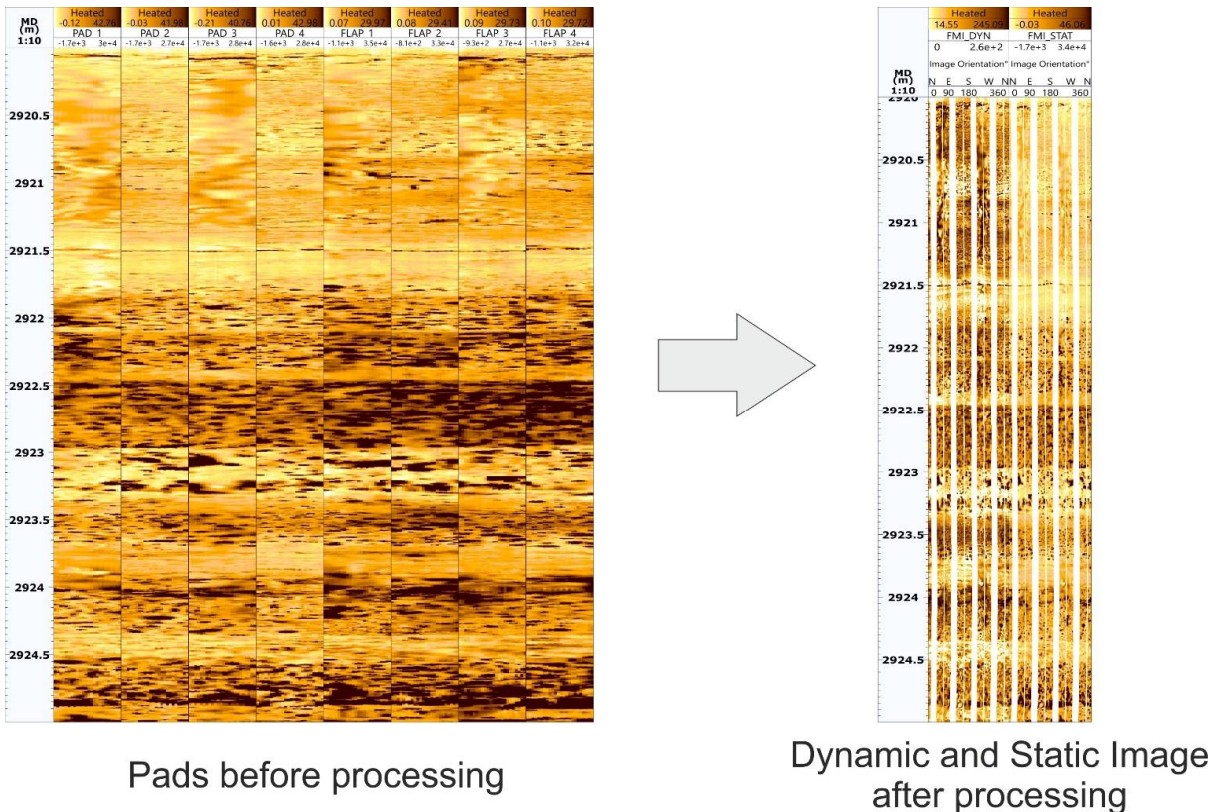

**Figure 5.** Results of the FMI processing for well A. For static image derivation, each value of resistivity or conductivity is mapped on a scale normalized for the entire interval. Dynamic normalization maps each value to a scale defined by the data range in a sliding window. Therefore, for the dynamic image (FMI_DYN), local contrast is enhanced. Note the increase in resolution on the dynamically processed images, especially above 2921.5 m. Resistive intervals are represented by light yellow shades. Darker shades and spots show the presence of possible pore space, where conductive mud filtrate can enter.

The processed FMI images yielded a continuous, high-resolution vertical image of the borehole interior, which was color-coded for its resistivity value. Typically, the FMI color scale corresponds to a range of resistivity, from conductive (dark colors) to resistive (light colors). The borehole image data can be processed to produce either "static" or "dynamic" images. For static images, the color scale range remains constant for the whole well. This allows the large-scale changes in resistivity to be identified, and a given color will represent the same resistivity value wherever it is in the well. However, subtle variations within a thin interval may not be seen as significant color changes. For dynamic images, the full range of colors is used for a thinner window (in this case 2 m). Because the range of resistivity in any interval is likely to be less than for the whole logged interval, more subtle variations in resistivity will be highlighted [32]. Low resistivity (dark colors, such as dark brown) normally represent areas occupied by water-based drilling mud, and include large pores, open fractures etc., whereas light colors (e.g., bright yellow) indicate high resistivity, which may be due to, for example, cemented fractures. Inclined planar features are imaged as sine waves [33].

### 3.3. Data Interpretation

Since the FMI data were processed, the next stage, the interpretation stage, became available. FMI interpretation can be divided into two types: qualitative and quantitative. Each is described below in more detail.

### 3.3.1. Structural and Textural FMI Analysis

Qualitative, manual interpretation of geological features, such as bedding structure, cross bedding, and lamination, was carried out on Techlog, Schlumberger. The structural

analysis described the changes in structural–textural objects (bedding structure) with depth. The classification of elements included the identification of bed boundaries, conductive and non-conductive fractures, and stylolites. Bed boundaries mark changes in the general nature of the formation, which depends on the conditions of sedimentation and the depositional history of the territory. Fractures are defined as rock discontinuities without visible displacement of rocks along the surface of the rupture. Stylolites are spiky, jagged lines or seams formed by the heterogeneous dissolution of rocks under the influence of overburden or tectonic pressure. Figure 6 shows examples of the FMI textures interpreted for the studied field N. The textures were used to conduct a manual typing of the FMI data since an understanding of sedimentary features is significant for the detection of reservoir geometry and petrophysical parameters.

## Structural/Textural features on FMI images

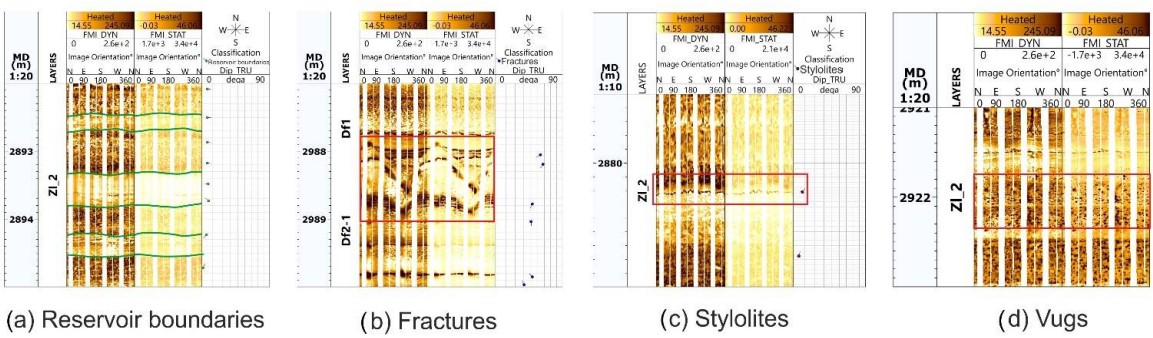

**Figure 6.** Example of features observed during the FMI interpretation. The green lines in (**a**) mark the boundaries between discrete layers. In (**b**), an open, conductive fracture is visible as a dark-colored sine wave between 2988 m and 2989 m, and an irregular, sub-horizontal stylolite is visible in (**c**). In image (**d**), the irregular dark patches are isolated vugs.

### 3.3.2. Quantitative Analysis of Vug Fraction (PoroSpect Method)

The interpretations described above were of a more qualitative nature. In order to evaluate the FMI data quantitatively, it was decided to apply the PoroSpect method, which allows for the assessment of vug fraction in the section. This decision was due to the fact that many productive carbonate formations have a complex dual-porosity system consisting of matrix (primary and secondary) porosity and fractures. The secondary porosity might contain vugs. Borehole electrical conductivity and resistivity images provide both small-scale resolution and azimuthal coverage to quantify the heterogeneous nature of the carbonate porosity component [27]. The analysis of the size and distribution of vugs consisted of the qualitative allocation of vug intervals and the calculation of the vug fraction in such intervals. The identification of the vuggy intervals was based on the presence of a spotted texture on the image logs. Dark, low-resistivity patches were interpreted as isolated pores filled by conductive drilling mud, and surrounded by low-porosity and higher-resistance matrix. The inferred pores were interpreted as secondary pores, or vugs, produced by local dissolution of components of the original carbonate rocks. Generally, when an image window is selected, the traditional Archie Equation (1) is used to calculate porosity from each image within the image window. The calculated porosity distribution provides insight into pore structure [34]. The estimation of vug fraction in the total porosity distribution was done by following the PoroSpect method (i.e., a porosity spectrum analysis based on borehole electrical images), which adopts the Archie–Dakhnov equation for the flushed zone, converted for the FMI data.

$$S_{xo}^n = \frac{aR_{mf}}{\varnothing^m R_{xo}}, \tag{1}$$

where $S_{xo}$ is the brine saturation of the flushed zone, $R_{mf}$ is the resistivity of the mud filtrate, $R_{xo}$ is the resistivity of the flushed zone, $\varnothing$ is the porosity, $a$ depends on the tortuosity, $m$ is the cementation factor, and $n$ is the saturation exponent.

Newberry et al. [35] obtained Equation (2) by assuming that $S_{xo}$ = 1.0, $a$ = 1.0, and $m$ = $n$ = 2.0 with account for variations in water saturation:

$$\varnothing_i = \varnothing_{ext}(R_{ext}C_i)^{\frac{1}{m}}, \tag{2}$$

where $\varnothing_i$ is the porosity at each borehole image electrode; $\varnothing_{ext}$ and $R_{ext}$ are the porosity and the shallow resistivity, respectively, from conventional logs; and $C_i$ is the conductivity of each button from the image [36].

Applying the principle shown in Figure 7, the presence of 192 buttons within the circumference of the well bore provided 192 values of porosity at each depth, rather than one discrete value. By analyzing the distribution of porosity and applying a cut-off value, it is possible to distinguish the matrix porosity from the vug fraction. In this research, the calculation of the percentage of vugs was based on the SDR cut-off method, considering a vertical step of 2 inches (50 mm), the resistance curve (GZ3), and the total porosity (PHIT; neutron density). This method locates the threshold at a user-defined fixed percentage (default value: 15%) above the mean porosity. Core observations and measurements provide a way to calibrate this value [27,37]. These qualitative data of the vug fraction were used further to distinguish between the different rock types.

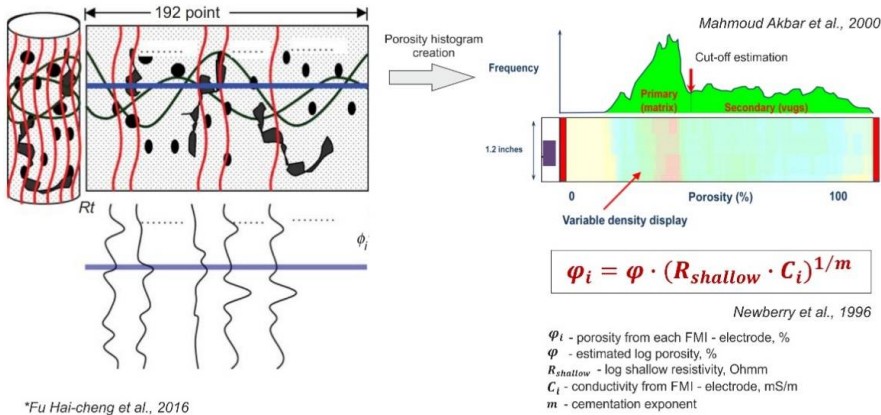

**Figure 7.** Calculation of the porosity spectrum from borehole images. Top right panel: x-axis = porosity distribution; y-axis = frequency. Bottom left panel: sketch map of the button-electrode conductive curves. Top left panel: the dimension of the figure is 192 points in the FMI. Black dots: vugs, dark green sinusoids: fractures, red vertical lines: button-electrode conductive curves, dark blue horizontal line: reference level for porosity histogram calculation, which is estimated at each depth of the FMI survey.

The final stage referred to rock differentiation by means of FMI (electrotyping of FMI), which combines qualitative and quantitative descriptions and core photos (Figure 8). The following FMI rock types were identified within D3zv, the Zavolzhsky horizon:

- Electrotype 1, characterized by a uniform and smooth FMI image, dark grey homogeneous limestones with partially uniform hydrocarbon luminescence in ultraviolet light, and high values of vug fraction;
- Electrotype 2, characterized by a patchy heterogeneous FMI, greenish limestones without hydrocarbon luminescence in ultraviolet light, low porosity, and the absence of vugs and presence of stylolites;
- Electrotype 3, characterized by a homogeneous and dark FMI showing large, lightly colored spots, limestones with large inclusions, clearly visible outlines of skeletal fragments, patchy hydrocarbon luminescence in ultraviolet light, and moderate values of vug fraction.

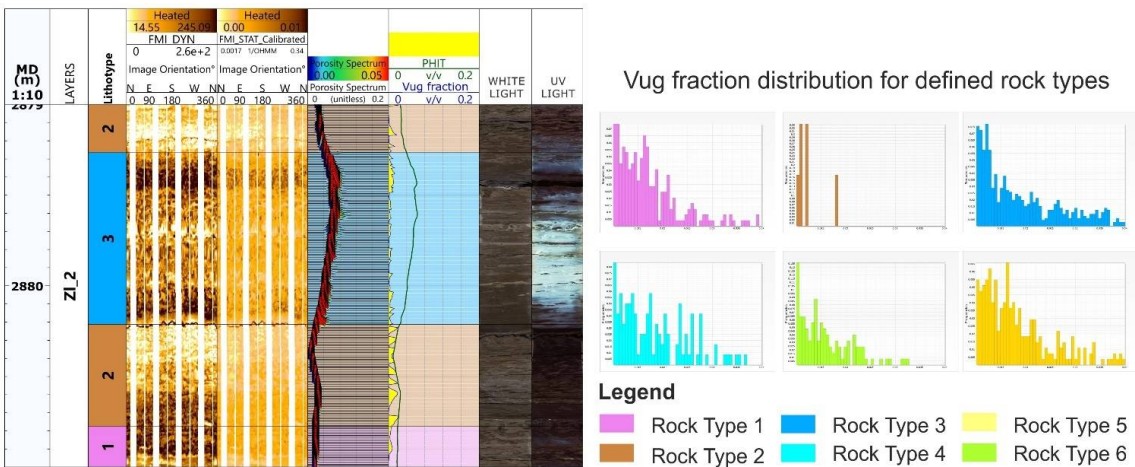

**Figure 8.** Vug fraction histogram with corresponding well-log template. Total porosity, PHIT, is derived from well logs. Distribution of vug fraction is obtained for each rock type, which was defined using core, FMI, and log data.

An additional type was discovered within D3fm, the Upper Famennian horizon:

- Electrotype 4, characterized by a uniformly smooth FMI showing small spots; moreover, the core photographs differed from those (with smooth FMI) of the Zavolzhsky horizon types. Limestones were uniform and had a lighter color. Hydrocarbon luminescence varied from uniform to patchy type.
- Finally, two types were identified within D3fm, the Lower Famennian horizon:
- Electrotype 5, characterized by FMI and core photographs showing small fractures and light-colored inclusions. Hydrocarbon luminescence varied from uniform to patchy type;
- Electrotype 6, characterized by a uniform smooth FMI and core photographs showing light-colored inclusions, dark spots, and fractures. Hydrocarbon luminescence varied from uniform to patchy type. Open fractures also contained hydrocarbons.

Parallel to the FMI qualitative and quantitative interpretation, the core study and petrography analysis were conducted. For carbonate reservoirs, full-bore core description is less effective due to the massive structure; the thin section tends to be more informative. This technique is based on the availability of automated image analysis systems that extract quantitative data on pore size and shape from thin section images in a relatively short amount of time [38].

### 3.3.3. Core Thin Section Analyses

The sedimentary components, structure–texture analysis, grain types, presence of detrital material, and diagenetic processes define lithofacies types. By examining the core photographs and thin sections, the previously described rock types were described:

- Lithofacies type 1: detrital limestone with individual algal fragments, dominated by dissolution and showing high values of total porosity and vug fraction;
- Lithofacies type 2: limestone composed of shell and algal detrital material with a dense micritic texture and stylolites, relatively low total porosity, and no dissolution;
- Lithofacies type 3: limestone with intraclasts (fragments), algal stromatoporoids, and skeletal remains that can have microfractures within the intraclasts and among them, and dissolution and recrystallization processes present;
- Lithofacies type 4: ooid-pelletic homogeneous limestone with high total porosity, which is mainly intergranular and associated with primary properties. It is represented by the similar behavior of porosity, permeability, and grain-fraction curves from petrography data (Figure 8);
- Lithofacies type 5: detrital limestone with algal fragments and ooids, with average values of total porosity and controlled by dissolution and low intensity of dolomitiza-

tion processes. Recrystallization is also present. This type is similar to type 1 but is common in the Lower Famenian;

- Lithofacies type 6: secondary crystalline dolomitized limestones and dolomites with microcracks clearly visible in thin section.

### 3.3.4. Petrographic Analysis

Petrographic analysis, in this case on limestones, dolomites, and associated deposits, is carried out with optical or electron microscopes. Petrography is an especially powerful tool because it enables the identification of constituent grains and crystals, the detailed classification of sediments and rocks, the interpretation of environments of deposition, and the determination of the complex history of post-depositional alteration (diagenesis) [39].

Petrographic data from wells were used to better justify the above core rock-typing (RT). Intervals characterized by a predominance of secondary processes were identified, and the types described in the previous sections were compared based on the following:

- the grain composition;
- the matrix and cement;
- the components of bioclasts and algal remains;
- diagenetic processes.

Based on the above, diagrams were produced to show the distribution of components and diagenetic properties in the studied reservoirs (Figure 9).

The dissolution processes observed in lithofacies type 1 occurred also in lithofacies types 3 and 6. The comparison between the porosity/permeability and amount of grains can help determine the prevailing factors of primary and secondary porosity. In the present study, the similar behavior of porosity, permeability, and grain fraction suggested a high intergranular primary porosity for lithofacies type 4.

Histograms of the wireline log values (i.e., gamma-ray log (GR), neutron log (NGR), density log (RHOB), acoustic log (DT), and laterolog (LLD)) were constructed for each lithotype (Figure 10); afterwards, typical values were selected for each lithotype with the intent of reconstructing a block curve of lithotypes using a prescribed script in Python.

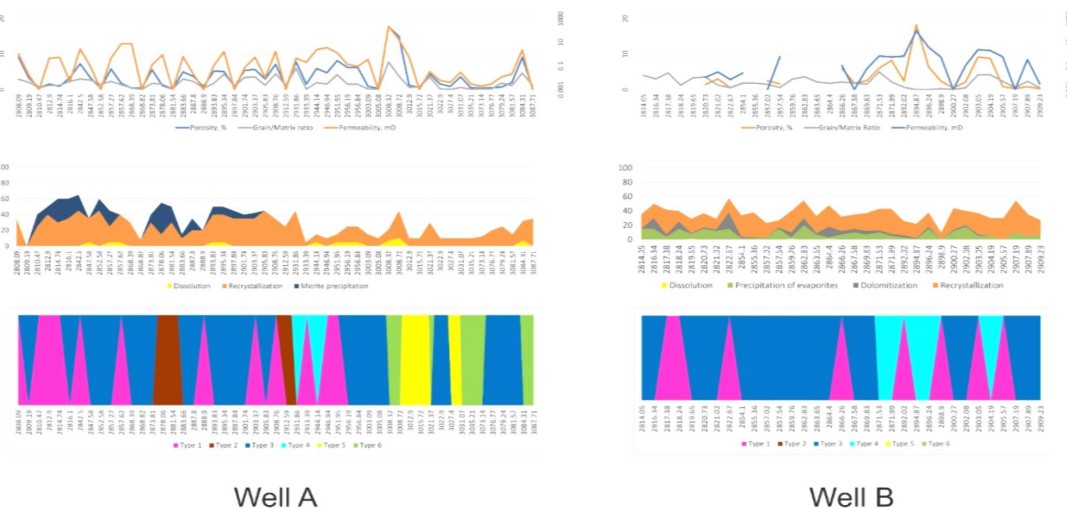

**Figure 9.** Stratigraphic distribution of lithotype diagenesis and reservoir properties for two wells, A and B. For well A, porosity and permeability values directly follow the changes in grain/matrix ratio, and dissolution and recrystallization emphasize the reservoir properties (e.g., at 3008.3 m). In well B, there is no visible dissolution, but dolomitization and the precipitation of evaporites occur through most of the cored interval. These diagenetic processes can either increase or reduce porosity and permeability. Both porosity and permeability are lower in the upper half of the cored interval, where evaporites are more common. Note also the increased porosity and permeability values at 2894.8 m, where the grain/matrix ratio is relatively low. It can be connected with smaller values of both evaporite precipitation and recrystallization.

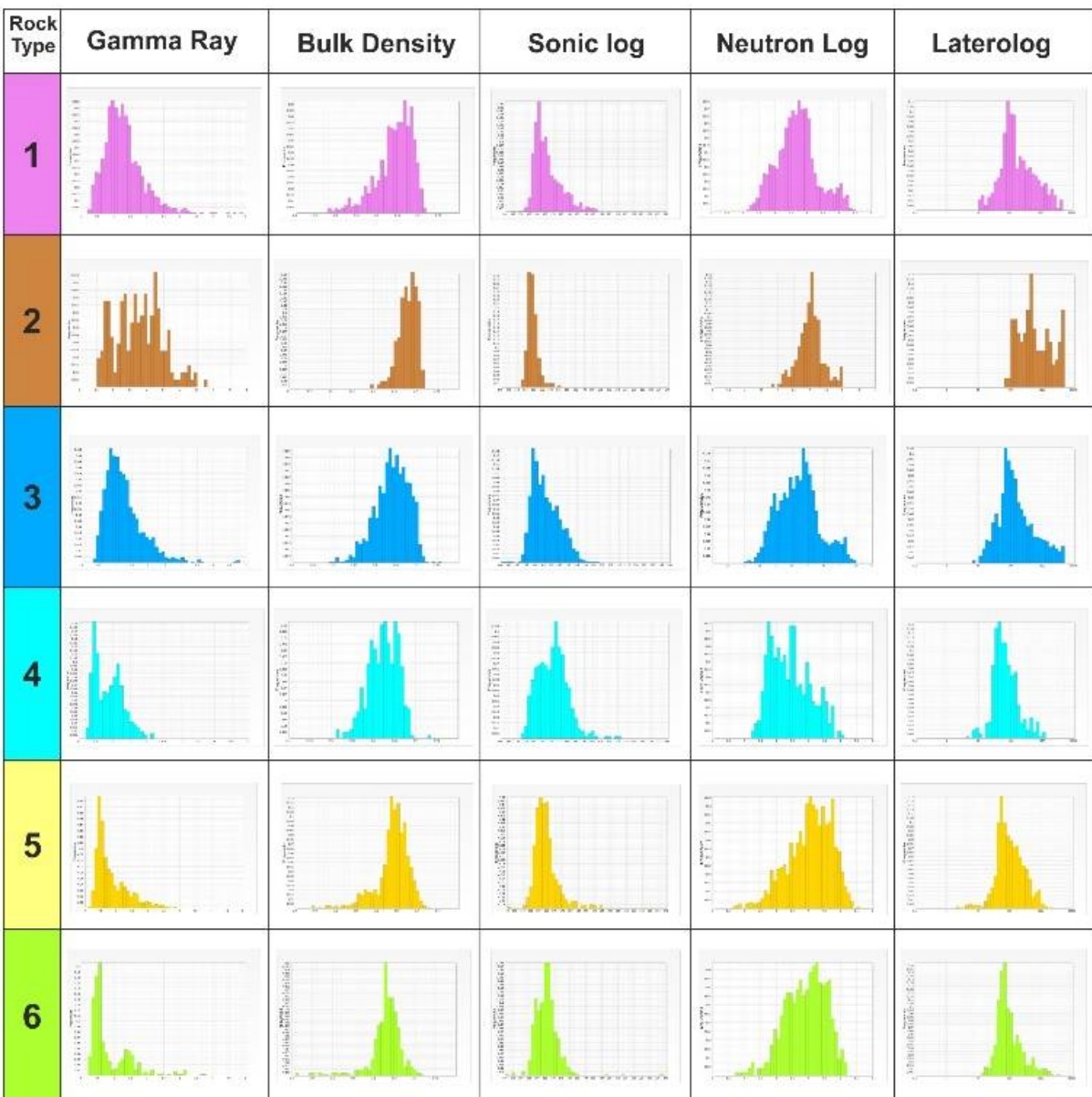

**Figure 10.** Histograms showing the distribution of the wireline log values: pink = lithofacies type 1, brown = lithofacies type 2, blue = lithofacies type 3, turquoise = lithofacies type 4, yellow = lithofacies type 5, and green = lithofacies type 6.

## 4. Results and Discussion

The proposed workflow consists of two parallel analyses. One is based on the study of FMI data, the other on the description of core, thin sections, and petrography. The results showed that in both cases, six types of rocks could be distinguished. In order to get the resulting rock type, which will be used later in geomodelling, it is necessary to find a connection between the distinguished electro and lithofacies types. This link consisted in identifying characteristic features of electro and lithofacies types.

Finally, six main rock types were differentiated along the studied section and a catalogue of rock types was compiled. This included, for each type, a distinctive FMI, a core photograph, a thin section photograph, the k–φ relationship, and the main values of the wireline log curves (Table 1).

**Table 1.** Summary table of the six main rock types with distinctive FMI images and their distinctive properties.

| Rock Type | FMI | | Core | | Thin Section | $k$–$\varphi$ Relationship | Vug Fraction, % | phi, % | perm, mD | Typical Well Log Values | | | | |
| | Static | Dynamic | DL | UV | | | | | | GR mR/h | NGR UE | RHOB, g/cm³ | DT mS/m | LLD Ohmm |
|---|---|---|---|---|---|---|---|---|---|---|---|---|---|---|
| 1 | | | | | | | 0.92 ± 0.6 | 4.35 ± 1.7 | 0.136 | 1.28 ± 0.45 | 3.22 ± 0.47 | 2.64 ± 0.03 | 166.2 ± 6.68 | 430 |
| 2 | | | | | | | 0.58 ± 0.3 | 2.38 ± 0.8 | 0.065 | 1.78 ± 0.58 | 3.8 ± 0.29 | 2.68 ± 0.02 | 159.2 ± 2.38 | 1147 |
| 3 | | | | | | | 1.13 ± 0.8 | 4.58 ± 1.7 | 0.14 | 1.24 ± 0.43 | 3.12 ± 0.48 | 2.63 ± 0.03 | 167 ± 6.6 | 418 |
| 4 | | | | | | | 1.05 ± 0.9 | 5.6 ± 1.7 | 0.24 | 0.87 ± 0.35 | 2.83 ± 0.37 | 2.62 ± 0.03 | 171 ± 7.11 | 111 |
| 5 | | | | | | | 0.72 ± 0.7 | 4.7 ± 1.7 | 0.09 | 0.83 ± 0.4 | 3.03 ± 0.37 | 2.64 ± 0.03 | 164.5 ± 5.3 | 171 |
| 6 | | | | | | | 0.71 ± 0.5 | 5.24 ± 1.7 | 0.114 | 0.84 ± 0.47 | 2.92 ± 0.29 | 2.92 ± 0.29 | 165.6 ± 5.07 | 170 |

For the Rock Type Catalogue, a Python script was created with the aim of defining rock types for wells with no core or FMI data (e.g., old exploration wells or production wells). The gamma-ray log, neutron log, and porosity values were taken as the basis of this classification; other logs had a supportive role. After the interpretation, well correlation was undertaken (see Figure 11). It should be noted that wells C and D had only conventional logs and core data (two test wells), whereas well B (one of the four training wells) had an extended range of surveys.

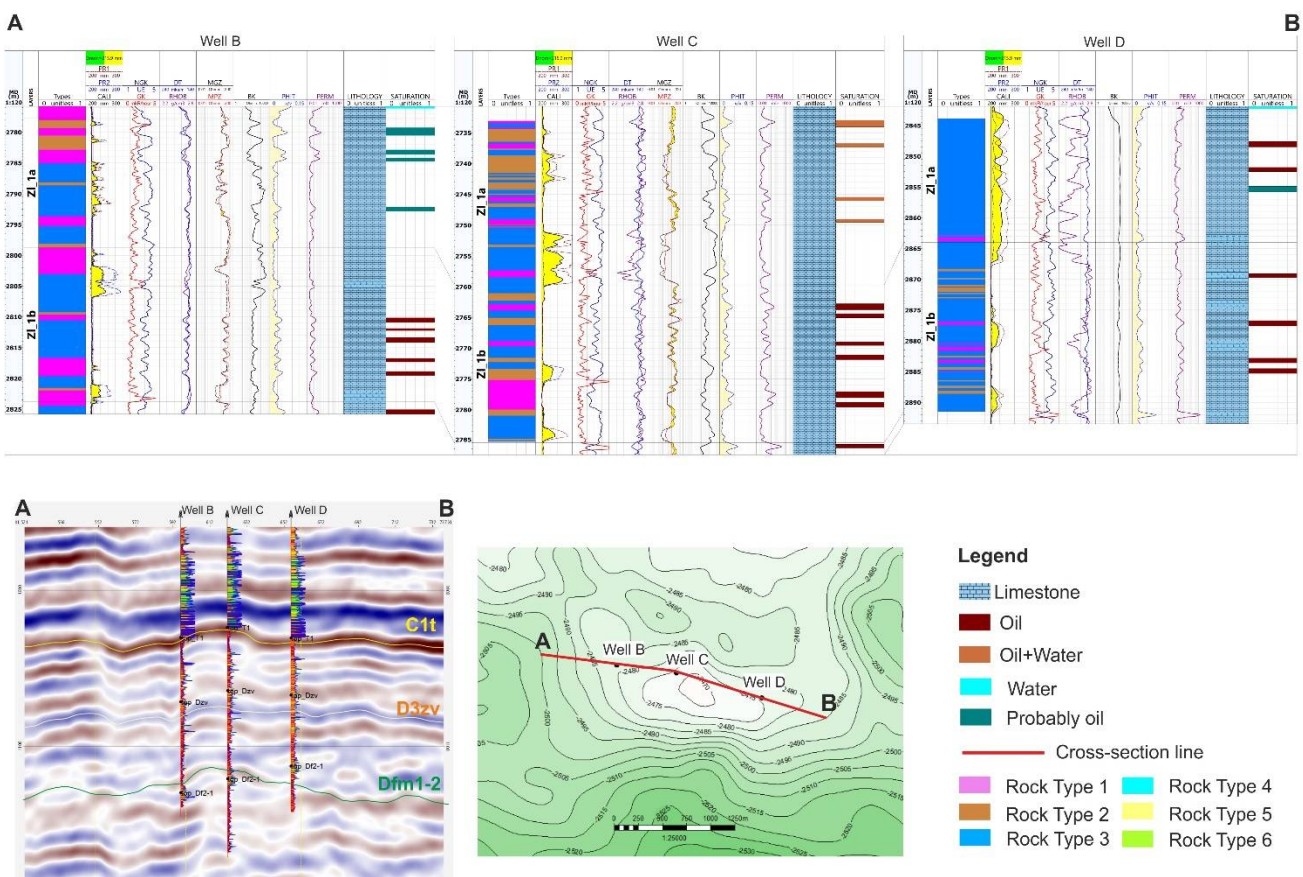

**Figure 11.** Final correlation of the wells based on the proposed rock-typing approach. A,B is cross section line through well B, C, D. For well B, rock types were assigned using the full dataset (FMI, petrography, etc.). For wells C and D, where only conventional logs and core were presented, rock types were created by dint of the developed Python script. The script is based on the Rock Type Catalogue.

The data obtained for the reservoirs were also matched with those of the different lithotypes in well C and well B. Notably, well D (Zl1a) was completely assigned to rock type 3. However, the values of log porosity and permeability were very low. This could be explained by the fact that the well is located within the shelf zone, particularly in the inner part of the small bioherm, where precipitation of evaporites is present. The same behavior was observed for well B. For well C, the most part of oil layers with higher values of porosity and permeability were correlated to rock type 1, which can possibly be explained by the higher values of vug fraction, which are typical for this type.

Figure 11 shows that rock types that are highlighted using the full logging suite with FMI data are more geologically reliable because they are based on the structural–textural analysis of borehole images. The definition of rock types for wells C and D was simply based on the existing values from conventional logs, which were identified from the Rock Type Catalogue. Since the number of special logs is usually limited due to economic

reasons, it increased the significance of the proposed methodology for geologists because it makes the inter-well correlation of rock types possible.

Rock type 1 is illustrated in Table 1, row 1, and in Table 2.

**Table 2.** Description of main reservoir parameters for RT1.

| Parameter | Description |
|---|---|
| Lithology | Detrital limestone with individual algal fragments |
| Petrophysics | Moderate porosity, which is controlled by the primary texture and the secondary process of dissolution |
| Reservoir/Non-reservoir | In most cases, can be the reservoir |
| FMI identification | Uniform and smooth |
| Typical features | The predominance of dissolution (high values of vug fraction); distributed throughout the section, but mainly within the D3zv |
| Dunham classification | Grainstone |
| Distinctive feature | High vug fraction |

Rock type 2 is illustrated in Table 1, row 1, and in Table 3.

**Table 3.** Description of main reservoir parameters for RT2.

| Parameter | Description |
|---|---|
| Lithology | Limestones composed of shell and algal detrital material |
| Petrophysics | Low porosity due to a dense micritic texture and the absence of dissolution |
| Reservoir/Non-reservoir | Non-reservoir |
| FMI identification | Heterogeneous dynamic image with spots showing different textures; the prevalence of high static resistance (light colors) and dark conductive spots (dark colors). |
| Typical features | High density due to a high micrite content that enhances the compaction process occurring in the primary granular limestone; numerous stylolites |
| Dunham classification | Wackestone |
| Distinctive feature | Stylolites |

Rock type 3 is illustrated in Table 1, row 1, and in Table 4.

**Table 4.** Description of main reservoir parameters for RT3.

| Parameter | Description |
|---|---|
| Lithology | Limestone composed of intraclasts and algal stromatoporoids accompanied by skeletal remains and microfractures |
| Petrophysics | Porosity controlled by the number of intraclasts, their composition, and diagenetic transformations (i.e., dissolution, recrystallization, evaporite precipitation) |
| Reservoir/Non-reservoir | Both reservoir and non-reservoir (due to combination of evaporite precipitation and dissolution) |
| FMI identification | Dynamic image clearly showing large bioclastic fragments. The static image shows shades of yellowish orange. |
| Typical features | Can have microfractures within the intraclasts and among them |
| Dunham classification | Rudstone |
| Distinctive feature | Large clasts in core |

Rock type 4 is illustrated in Table 1, row 1, and in Table 5.

**Table 5.** Description of main reservoir parameters for RT4.

| Parameter | Description |
|---|---|
| Lithology | Homogeneous limestone composed of ooids/pellets |
| Petrophysics | High values of intergranular primary porosity and good k–φ correlation. Secondary dissolution and dolomitization processes lead to a relatively high porosity. |
| Reservoir/Non-reservoir | Mostly reservoir |
| FMI identification | Homogeneous and smooth image showing finely selected spotting. Distributed only within the Upper Famennian horizon. |
| Typical features | Figure 8, permeability (red), porosity (blue), and granularity (green) curves show similar trends, indicating the prevalence of primary intergranular porosity. |
| Dunham classification | - |
| Distinctive feature | Intergranular primary porosity |

Rock type 5 is illustrated in Table 1, row 1, and in Table 6.

**Table 6.** Description of main reservoir parameters for RT5.

| Parameter | Description |
|---|---|
| Lithology | Detrital limestone with algae fragments and ooids |
| Petrophysics | Porosity controlled by dissolution, dolomitization, and recrystallization |
| Reservoir/Non-reservoir | Potential reservoir |
| FMI identification | Homogeneous and smooth image showing insignificant inclusions |
| Typical features | It is similar to rock type 1, but specific to the Lower Famennian |
| Dunham classification | Packstone |
| Distinctive feature | Wavy texture on a dynamic image, tectonic fractures |

Rock type 6 is illustrated in Table 1, row 1, and in Table 7.

**Table 7.** Description of main reservoir parameters for RT6.

| Parameter | Description |
|---|---|
| Lithology | Secondary crystalline dolomitized limestones and dolomites |
| Petrophysics | Porosity resulting from primary uniformity and subsequent dolomitization, which contributed to the development of microcracks. Intercrystalline porosity is presented. |
| Reservoir/Non-reservoir | Fractures can contain hydrocarbons. |
| FMI identification | Image showing numerous fractures |
| Typical features | Fractured and specific to the Lower Famennian |
| Dunham classification | - |
| Distinctive feature | Fractures |

## 5. Conclusions

The proposed workflow is valuable in carbonate fields with limited data on core extracted and FMI. It can help geologists describe the reservoirs rocks in the wells with conventional logs. However, a borehole imager is mandatory, with at least one in the field or study area. If the conditions do not allow the borehole microimage to be recorded, or its interpretation is unsatisfactory, the approach is not applicable.

Identification of the rock types is a vital step for complex geological systems when it is difficult to characterize the reservoir. This is especially true for carbonate systems, as the relationship between facies, porosity, and permeability is less systematic than in clastic successions. The problems are exacerbated by the limited number of wells with complete datasets, as is usually the case. In addition, in Russia, the development of onshore fields, especially at shallow depths, can be economically viable in reservoirs with low porosity and permeability values, which, for example, is inappropriate in offshore conditions. The low porosity and permeability of many fields make identifying more than average sweet spots vitally better for field development. Thus, the value of the approach grows with possibility of using it in new wells, without core.

The proposed integrated rock-typing approach is based on formation microimaging (FMI) and core data. FMI analysis is an excellent method for rock-typing due to its high vertical and horizontal resolution and ability to distinguish thin layers with different textures and structures and correlate them with core data. In wells with complete log sets, in addition to core and FMI, wireline response for different rock types can be established using Python script.

Moreover, the workability of the workflow is shown as the differentiation of six rock types in the real field N, using FMI images, petrophysical relationships, and petrographic data. Using FMI images and core data (including microscopic data), it is possible to establish sedimentary facies, and identify and quantify dissolution (including caverns), styolites, fractures, recrystallization, evaporite deposition, and dolomitization, all of which influence porosity and permeability.

The analysis of such wells allows for the creation of a comprehensive log interpretation methodology that can be applied to other wells with only conventional logs. It is important to keep in mind the difference in resolution between FMI and conventional logs. Using various clustering analysis approaches that could be used on well logging and rock physics data plays an important role here. Methods such as k-means, model-based, hierarchical clustering, and fuzzy clustering are used to identify groups of similar objects in multivariate data sets collected from fields automatically. The rock types provided by the proposed

method, each with its own relationships, help in the creation of reliable geological models of carbonate reservoirs and their populations with petrophysical properties, especially for reservoirs with limited data available. This method allows core and log data to be linked at different scales through FMI log data, and allows for cross-well correlation of rock types. Rock-type distribution maps can be generated to accurately target net productivity targets and provide a more reliable drilling plan. This should lead to more successful development of carbonate reservoirs with low porosity and permeability values.

**Author Contributions:** Investigation, D.C., P.K., and E.M.; resources, E.M. and M.T.; data curation, A.G., V.R., P.K., E.M., and M.T.; writing—original draft preparation, D.C., P.K., and E.M.; writing—review and editing, I.C., E.M., and P.K.; funding acquisition, D.C., P.K., and E.M. All authors have read and agreed to the published version of the manuscript.

**Funding:** This research was supported by Tomsk Polytechnic University Competitiveness Enhancement Program.

**Institutional Review Board Statement:** Not applicable.

**Informed Consent Statement:** Not applicable.

**Data Availability Statement:** Not applicable.

**Acknowledgments:** The authors gratefully acknowledge the support from the Gazpromneft Science & Technology Centre.

**Conflicts of Interest:** The authors declare no conflict of interest. The funders had no role in the design of the study; in the collection, analyses, or interpretation of data; in the writing of the manuscript, or in the decision to publish the results.

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
