# Peer review of "An Integrated Approach for Formation Micro-Image Rock Typing Based on Petrography Data: A Case Study in Shallow Marine Carbonates"

_geosciences, doi:10.3390/geosciences11060235_

Round 1

Reviewer 1 Report

Review on Geosciences-1229603

The paper is well written and organized. Minor revisions are required.

  1. Lines 41-43: Citations are required.
  2. The literature review needs to be expanded to provide a full paragraph should be added (May be a second paragraph in the Introduction section) to provide review about using the various clustering analyses approaches that have been using on well logging and rock physics data. The main clustering analyses that should be reviewed are k-means, model-based, and hierarchal clustering methods. You may consider the most recent and popular reference for the review:

McCreery, E. and Al-Mudhafar, W. Geostatistical Classification of Lithology Using Partitioning Algorithms on Well Log Data - A Case Study in Forest Hill Oil Field, East Texas Basin. 79th EAGE Conference and Exhibition, At Paris, France. https://doi.org/10.3997/2214-4609.201700905

Tang H, White C, Zeng X, Gani M, Bhattacharya J Comparison of multivariate statistical algorithms for wireline log facies classification. AAPG Ann Meet Abstr 88:13 (2004).

Al-Mudhafar, W.J. Integrating well log interpretations for lithofacies classification and permeability modeling through advanced machine learning algorithms. J Petrol Explor Prod Technol (2017) 7: 1023. https://doi.org/10.1007/s13202-017-0360-0

Pirrone, M., Battigelli, A., & Ruvo, L. (2014, October 27). Lithofacies Classification of Thin Layered Turbidite Reservoirs Through the Integration of Core Data and Dielectric Dispersion Log Measurements. Society of Petroleum Engineers. doi:10.2118/170748-MS

Al-Mudhafar, W., Bondarenko, M. (2015). Integrating K-Means Clustering Analysis and Generalized Additive Model for Efficient Reservoir Characterization. The 77th EAGE Conference & Exhibition Incorporating SPE EUROPIC, Madrid, Spain. 
 https://doi.org/10.3997/2214-4609.201413024

Avseth P, Mukerji T (2002) Seismic lithofacies classification from well logs using statistical rock physics. Petrophysics 43(02):70–81. https://doi.org/10.2118/170748-MS

Al-Mudhafar, W. J., Al Lawe, E. M., & Noshi, C. I. (2019). Clustering Analysis for Improved Characterization of Carbonate Reservoirs in a Southern Iraqi Oil Field. Offshore Technology Conference. Houston, Texas. https://doi.org/10.4043/29269-MS

Euzen, T., Delamaide, E., Feuchtwanger, T., & Kingsmith, K. D. (2010). Well Log Cluster Analysis: An Innovative Tool for Unconventional Exploration. Canadian Unconventional Resources and International Petroleum Conference, Calgary, Alberta, Canada. https://doi.org/10.2118/137822-MS

  1. The last paragraph in the introduction section should mention to the borehole image and the advantages of the integrated rock-typing approach and how it differs from other conventional approaches or ML methodologies.
  2. Again you should explain the advantages and disadvantages of the new integrated approach in section 2 (Materials and Methods) and state how it differs from other conventional and ML methods.
  3. Section 3 and 4 look very good and no further revisions are required in them.
  4. Conclusions may be improved considering the commented points above.
  5. References should be updating considering the suggested references and may be others.

Reviewer 2 Report

I have pleasure to read the article by Kharitontseva P. et al.  The paper deals with a important subject of Rock typing for carbonate reservoir rocks and the approach of integrating various data set of RCA, SCAL, FMI, petrography et. al  is the right way. The study area is of very interesting for readers. The paper is well structured and clearly written. I would like to recommend it to be published in MDPI Geosciences.

Author Response

Thank you very much for your comments and suggestions